# Variations in the Second Derivative of a Photoplethysmogram with Age in Healthy Korean Adults

**DOI:** 10.3390/ijerph20010236

**Published:** 2022-12-23

**Authors:** Jungmi Choi, Min-Goo Park

**Affiliations:** 1Human Anti-Aging Standards Research Institute, Uiryeong 52111, Republic of Korea; 2Department of Bioenvironmental Chemistry, Jeonbuk National University, Jeonju 54896, Republic of Korea

**Keywords:** second derivative wave, photoplethysmogram, aging index, arterial distensibility, healthy Koreans

## Abstract

Second derivative of photoplethysmogram (SDPTG) indices correlate with aging and vascular health. The trend of SDPTG indices with age has not yet been studied in the Korean population. Various SDPTG indices were measured in 300 healthy Koreans (150 men and 150 women), aged 19–69 years, stratified into five age groups consisting of 60 people (30 men and 30 women) in each age group from their 20s to 60s. The values of the SDPTG indices clearly showed distinct variations with age in healthy Korean groups (*p* < 0.001 for all indices). b/a increased linearly with age (y = 0.0045x − 0.803), as did SDPTG aging index (SDPTG-AI) (y = 0.0162x − 1.1389). c/a decreased linearly with age (y = −0.0044x + 0.1017), as did d/a (y = −0.0062x + 0.034) and e/a (y = −0.001x + 0.2002). A significant sex difference was shown in b/a and e/a ratios and SDPTG-AI (*p* < 0.001 for all indices); women had a higher b/a ratio (−0.55 ± 0.14 versus −0.65 ± 0.13) and SDPTG-AI (−0.33 ± 0.3 versus −0.52 ± 0.33) and a lower e/a ratio (0.13 ± 0.06 versus 0.18 ± 0.07) than men. A linear regression model of diverse SDPTG indices was provided according to the age of healthy Koreans, which may be valuable in preventing diseases related to vascular conditions by estimating the degradation of arterial function.

## 1. Introduction

As the aging population of the world grows at an exponential rate, it becomes a great challenge for public health care [1]. Therefore, the identification of reliable factors is necessary to prevent diseases related to aging [2]. Arterial conditions are a major risk factor for aging and may contribute to hypertension, atherosclerosis, diabetes mellitus, and end-stage renal disease [2,3,4,5].

Pulse wave analysis is a well-established method for evaluating vascular load and vascular aging and could thus be useful for assessing the vascular effects of hypertension, atherosclerosis, and aging [2,6,7]. Photoplethysmography (PTG) is a simple, non-invasive technique for measuring changes in blood volume [8,9,10,11]. Using this method, the pulse waveform reflecting arterial vessel elasticity can be evaluated. However, because it is difficult to visually grasp detailed information of shape, including the inflection point of the pulse, the second derivative of the finger photoplethysmogram (SDPTG) has been used as an objective and reliable method of measurement [7,12,13,14].

In particular, SDPTG indices are useful for evaluating arterial deterioration as they show a distinctive trend that increases or decreases linearly with age [7,15,16]. In 2017, the SDPTG was introduced as a new aging index [17], and in 2021, the author of this study suggested that the SDPTG aging index (SDPTG-AI), which involves performing measurements in the elderly with the same instrument and protocol as in this study but with detection on the ear lobe rather than the fingertip, showed a clear correlation with age in a multiple regression model [14].

SDPTG indices showed higher or lower values for women than for men in the same age group [7,13]. This is attributed to the fact that women have a faster pulse wave velocity than males because the arterial length and diameter in women are shorter and smaller, respectively than in men [18]. Therefore, SDPTG should be evaluated separately according to sex.

Studies on the SDPTG tendency with aging are concentrated in Japanese and Western populations [2,5,7,12,13,19]. Only a few studies have been conducted in South Korea. Furthermore, each SDPTG index could not be quantitatively predicted with functional age because the number of study participants for each age group was small and varied [16,17]. The Japanese SDPTG has presented quantitative SDPTG by sex and age [7]. However, considering the average height difference between Japanese and Koreans [20], it will most likely not be applicable to Koreans.

Therefore, we presented a linear regression model of SDPTG indices that changed with age in healthy Koreans by having a uniform number of study participants stratified by sex and age. We also provided the reference range for SDPTG by sex and age group.

## 2. Materials and Methods

### 2.1. Subjects

A total of 300 healthy volunteers were recruited from the Clinical Trial Centre of Asan Medical Centre (AMC, South Korea), stratified into five age groups consisting of 60 people each, between the age groups of 20s to 60s. Participants were selected based on the following inclusion criteria:Healthy Korean males and females aged between 19 and 69 yearsNon-smokersNon-pregnant womanSubjects with a BMI (body mass index) of 18 to 30 kg/m^2^Subjects who voluntarily decided to participate in this clinical trial for 2–3 h and gave their written consent

However, subjects with the following criteria were excluded from the study:Subjects who had consumed food other than water within 1 h of SDPTG measurementsSubjects who performed strenuous exercise within 1 h of SDPTG measurementsSubjects judged to be sleep deprived for less than 4 h the day before the measurementsSubjects who were expected to have a lot of hand movements because they could not control their movements when measuring the SDPTGSubjects who were taking drugs for the treatment of central nervous system or cardiovascular diseasesSubjects taking medications for the treatment of hyperthyroidism or hypothyroidismPatients suffering from diabetes for more than 10 years

This study was conducted as part of the quantitative photoplethysmogram (PTG) parameter statistical standardization for healthy Koreans, a brain frontier project promoted by the Ministry of Education. Written consent was obtained from each participant prior to study participation. The study protocol was approved by the Institutional Review Board (ASAN MEDICAL CENTER INSTITUTIONAL REVIEW BOARD, IRB number: AMC-IRB-2007-0305). After gathering basic demographic information, clinical research nurses examined the participants with the SDPTG. The study was conducted in accordance with the guidelines of the Declaration of Helsinki.

### 2.2. SDPTG Measurements

All subjects sat in a comfortable position with their eyes open, and their pulse waves were measured with a photoplethysmometer for 5 min [21]. A PTG sensor (Model: ubpulse T1 (Pulse Analyzer, KFDA Certification No. 11-1296), LAXTHA Inc., Daejeon, South Korea) was placed on the left index finger [21].

The hand with the sensor was placed on a table at the heart level, and nail polish or other foreign objects on the finger with the sensor were removed [22]. In addition, items that put pressure on the arms or fingers, such as sleeves, disposable bands, and rubber bands that fit the body, were removed. Caution was taken not to breathe deeply or abdominally during measurements to prevent respiratory sinus arrhythmias (RSA) [22].

With the device, PTG and second derivative PTG (SDPTG) measurements were performed simultaneously, as shown in Figure 1. Each SDPTG waveform was displayed after the derivative of the PTG in the device. The operator minimized data contamination by monitoring the signal shape and ensuring that it was not corrupted by sensor or finger movements.

### 2.3. Biomarkers and Calculations of SDPTG

The SDPTG is a waveform obtained by differentiating the PTG wave twice. It is easy to understand the detailed information obtained from the PTG shape, such as inflection points; the PTG wave, is also composed of several positive and negative peaks, as shown in Figure 1. Each peak of the SDPTG consists of waves a, b, c, and d, which correspond to the systolic wave and the diastolic e wave. They are defined as follows: (1) “a” wave is the large positive peak first appearing in an SDPTG signal, (2) “b” wave is the negative peak immediately following the a wave, (3) “c” wave is the positive peak immediately following the b wave, (4) “d” wave is the negative peak immediately following the c wave, and (5) “e” wave is the positive diastolic peak that appears immediately after the “d” wave.

As shown in Figure 1a–e waves represent the values corresponding to the amplitude of each peak, and the SDPTG waveform analysis was utilized by converting the relative values of b, c, d, and e waves based on the a wave. That is, b/a, c/a, d/a, and e/a ratios, which are the relative amplitude values of b, c, d, and e waves to the wave amplitude, were used as the SDPTG indicators in this study. The SDPTG-AI, the second derivative of the PTG aging index, was also used. This index reflects the functional aging level of the artery and was calculated as b/a-c/a-d/a-e/a. This is combined by setting the weights as +1 (proportional) and −1 (inversely proportional) to have a higher correlation as the level of functional aging of blood vessels increases, based on the fact that the b/a ratio is proportional to age, while the c/a, d/a, and e/a ratios are inversely proportional to age. Therefore, the SDPTG-AI is also commonly known as an early prediction index for arteriosclerosis, which is a representative condition of severe functional aging of blood vessels [7,23,24].

Arrhythmic events, ectopic beats, and noise effects may alter the estimation of the SDPTG [25]. Therefore, in our study, clean pulse data free of ectopy and noise were preferentially selected and used to extract SDPTG indices.

### 2.4. Statistical Analysis

Simple regression analysis was performed to reveal the trend of SDPTG indices according to the age of the subjects. An independent *t*-test was conducted to analyze the sex differences with respect to SDPTG parameters in each age group. Statistical analysis was performed using SPSS ver. 23 (SPSS Inc., Chicago, IL, USA, 2009). The significance level was set to α = 0.05 for all statistical tests (two-tailed).

## 3. Results

This study included a total of 300 participants. Before analysis, two experts screened the data to remove inappropriate samples that could not be analyzed due to measurement errors caused by finger movements or an undistinguished pattern of the signals, leaving 292 participants (147 men and 145 women) for analysis, as shown in Table 1. Detailed SDPTG indices on the participants are described in Appendix A.

Figure 2 shows PTG and SDPTG signals measured for five participants aged in their 20s, 30s, 40s, 50s, and 60s. The SDPTG signals of the participants demonstrate how the b wave becomes less negative and the d wave more negative with age. Accordingly, the SDPTG-AI and b/a increased to −1.01 and −0.9, respectively, in a participant aged 20s, to −0.51 and −0.71 in one aged 40s, and to −0.01 and −0.51 in one aged 60s. d/a decreased to −0.07 in a participant aged 20s, to −0.31 in one aged 40s, and to −0.49 in one aged 60s.

The values of the SDPTG indices (SDPTG-AI and b/a, c/a, d/a, e/a ratios) clearly showed a distinct trend with age in healthy Korean groups, as shown in Figure 3 (*p* < 0.001 for all indices). b/a increased linearly with age (y = 0.0045x − 0.803, R^2^ = 0.2064), as did SDPTG-AI (y = 0.0162x − 1.1389, R^2^ = 0.4812), while c/a decreased linearly with age (y = −0.0044x + 0.1017, R^2^ = 0.3119), as did d/a (y = −0.0062x + 0.034, R^2^ = 0.4792) and e/a (y = −0.001x + 0.2002, R^2^ = 0.0453).

Figure 4 shows age-dependent changes in the main SDPTG indices in both the male and female groups (*p* < 0.001 for all indices). b/a, a reflection of the aortal stiffness, increased with age in both the male (y = 0.0053x − 0.8863, R^2^ = 0.3533) and female groups (y = 0.0038x − 0.722, R^2^ = 0.1549). Similarly, the SDPTG-AI, an aging index of arteries, also increased in both the male (y = 0.0177x − 0.1329, R^2^ = 0.5729) and female groups (y = 0.0147x − 0.9794, R^2^ = 0.4823). d/a decreased with age in both the male (y = −0.0068x + 0.0738, R^2^ = 0.5088) and female groups (y = −0.0056x − 0.0049, R^2^ = 0.4834).

Table 1 shows the sex differences in the SDPTG indices. Females had a higher b/a ratio (−0.55 ± 0.14 versus −0.65 ± 0.13, *p* < 0.001), a higher SDPTG-AI (−0.33 ± 0.3 versus −0.52 ± 0.33, *p* < 0.001) and a lower the e/a ratio (0.13 ± 0.06 versus 0.18 ± 0.07, *p* < 0.001) than males.

## 4. Discussion

A linear regression model of diverse SDPTG indices has been presented, which considers the age of healthy Koreans, and reference ranges were provided for Korean females and males based on age groups. The SDPTG indices clearly showed an increase in b/a and SDPTG aging index and a decrease in c/a, d/a, and e/a with age. Each index exhibited a significant age-dependent trend (Figure 3). This is consistent with the trend reported in previous studies of Japanese and Western people. Takazawa et al., showed that the b/a ratio and SDPTG-AI rose with age, while the c/a, d/a, and e/a ratios declined with age in 600 Japanese individuals [7]. Bortolotto et al., found that the SDPTG-AI had a significant relationship with age, and was higher in participants aged > 60 years than in those aged < 60 years in 524 French individuals [2]. Although a regression model of SDPTG with age has not yet been provided, SDPTG parameters in Koreans also showed an age-dependent change consistent with those in Japanese people [16].

Among the SDPTG indices, the negative b/a ratio reflects the degree of aortic distensibility [6]. This ratio increases if the vascular wall is well-stretched with a low vascular load when the aortic blood volume is rapidly increased at the moment of cardiac output. Conversely, when the aortic vessel wall is stiff, this negative ratio decreases. This means that the b/a ratio increases with age, which enhances aortic stiffness, and reflects the vascular state of the aorta as the first vascular response when blood is released from the left ventricle [7,13]. A negative d/a index reflects arterial compliance or flexibility of the peripheral artery wall [13]. When there is functional vascular wall tension due to high blood pressure or when arterial flexibility is reduced due to arteriosclerosis, the magnitude of the reflex wave in the peripheral arteries increases, and accordingly, this negative index increases. Thus, the d/a ratio decreases with age [7,13]. c/a is a waveform which, along with changes in b and d waves, reflects decreased arterial stiffness and decreases with age [7,26]. The e wave reflects the initial rise of the diastolic wave that appears better in young and elastic vessels, and the e/a ratio tends to decrease slightly with age [7]. SDPTG-AI is calculated as (b-c-d-e)/a and reflects functional arterial aging [2,7]. It is commonly known as a parameter for assessing arteriosclerosis, which is representative of severe functional arterial aging. Several previous studies have also reported that patients with any history of arteriosclerotic diseases, including hypertension, diabetes mellitus, and hypercholesterolemia, had higher SDPTG-AI values [2,7,27].

In particular, this study has presented a decisive linear regression model of diverse SDPTG indices, such as b/a, d/a ratios, and SDPTG-AI in both male and female groups, in contrast with previous studies which provided an age-dependent trend regardless of sex (Figure 4). b/a, d/a, and SDPTG-AI were selected among the indices because b/a and SDPTG-AI are the most promising indices in the assessment of arterial health as well as aging [7,26], and because the magnitude of the slope of the trend line for the age of b/a, d/a, and SDPTG-AI was more extensive than that of c/a and e/a (Figure 2). The R^2^ values in the regression model that assessed the relationship between b/a, d/a, and SDPTG-AI with age for all participants were 0.2064, 0.4792, and 0.4812, respectively, whereas those for the male participants were 0.3533, 0.5088, and 0.5729, respectively and for the female participants were 0.1549, 0.4634, and 0.4823, respectively (Figure 4). The correlation between aging and each SDPTG index in the male participants was stronger than those of all participants but was lesser or equal to that of female participants. Thus, the functional age of each SDPTG parameter can be estimated quantitatively for each group (male or female). When the functional age calculated by this model is older than the actual age (i.e., 10 years older), they will be included in a group requiring health management. Therefore, arterial health and age-related diseases like atherosclerosis, diabetes, and end-stage renal disease could be prevented by recommending dietary management and exercise.

Only a few studies have reported sex differences in SDPTG indices. Women had a higher b/a ratio and SDPTG aging index and a lower e/a ratio than men in Japan [7]. Iketani et al., found that the SDPTG-AI declined with age from 3 to 18 years and then increased in young men, while it declined with age from 3 to 15 years and then increased in young women [13]. The entire Korean group in this study showed sex differences in the b/a, e/a, and SDPTG-AI (Table 1), which showed a consistent trend to that seen in Japanese people aged 30 to 80 years. This study is also the first to investigate all SDPTG indices for sex differences in the Korean population.

SDPTG indices were known to be affected by hemodynamic, autonomic, adiposity and emotional factors as well as ageing [2,7,28,29,30]. A recent study showed SDPTG indices had a significant relationship with those factors except emotional factors [31]. In this study, to avoid introducing hemodynamic, autonomic or adiposity effects to the SDPTG indices, healthy people without hypertension, obesity, or autonomic deterioration were recruited. In addition, SDPTG in the participants was calculated accurately only when nervous system drugs, caffeinated drinks, food, and exercise had not been administered 1 h before the assessment [32]

In conclusion, the distinct age-dependent trend of the SDPTG indices in this study will be valuable in preventing diseases associated with aging or arterial function degradation, by providing defining standards for its screening, with sensitive levels prior to typical symptoms related to arterial functional deterioration. Furthermore, the SDPTG is predicted to be used more actively to assess quantitative changes in sensitive levels before and after therapy or treatment, such as drugs, stimuli, and exercise.

## Figures and Tables

**Figure 1 ijerph-20-00236-f001:**
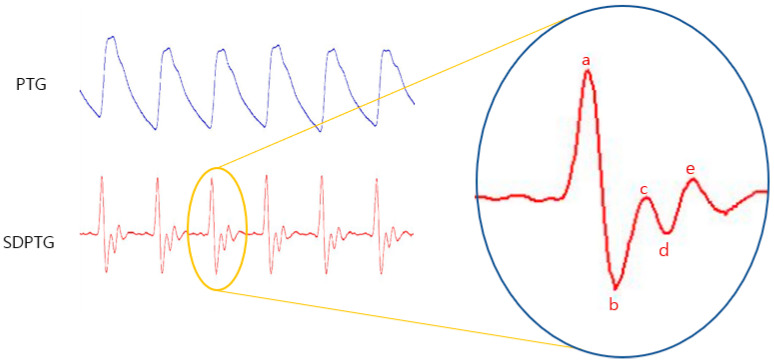
Photoplethysmogram (PTG) and second derivative PTG (SDPTG) signals.

**Figure 2 ijerph-20-00236-f002:**
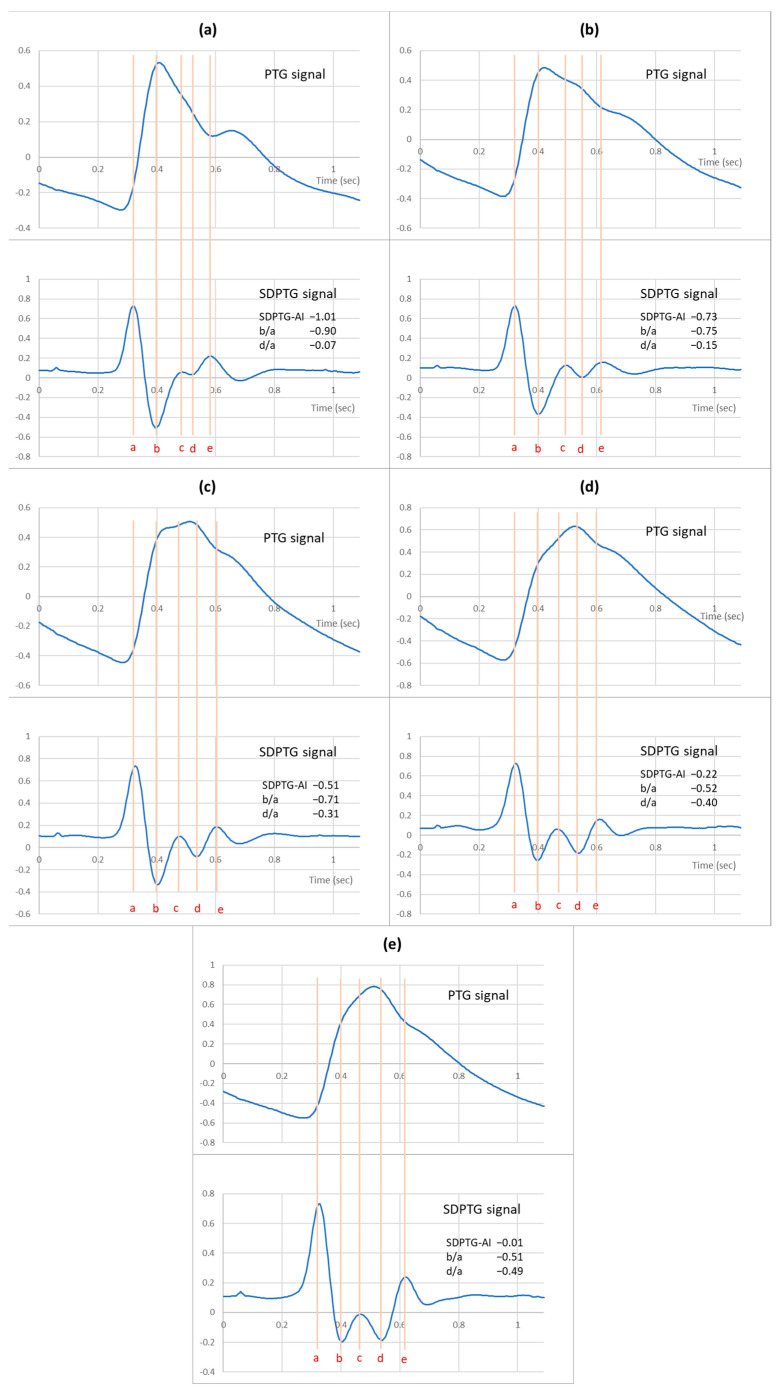
Photoplethysmogram (PTG) and second derivative PTG (SDPTG) measured for five participants aged in their 20s (**a**), 30s (**b**), 40s (**c**), 50s (**d**) and 60s (**e**).

**Figure 3 ijerph-20-00236-f003:**
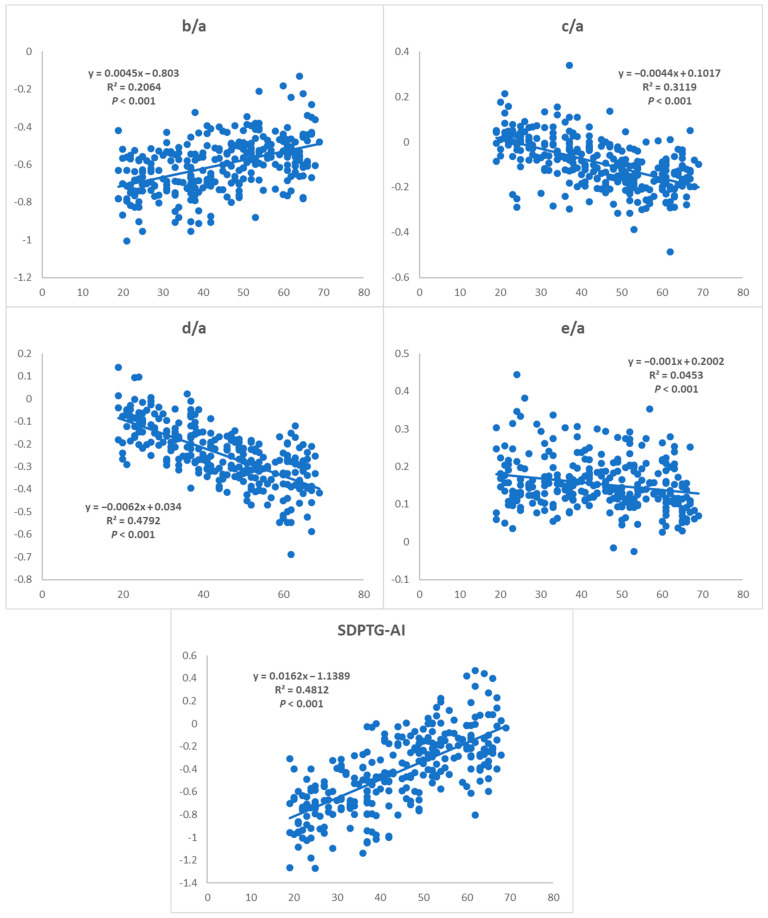
Variations in second derivative photoplethysmogram (SDPTG) indices with age in healthy Korean groups.

**Figure 4 ijerph-20-00236-f004:**
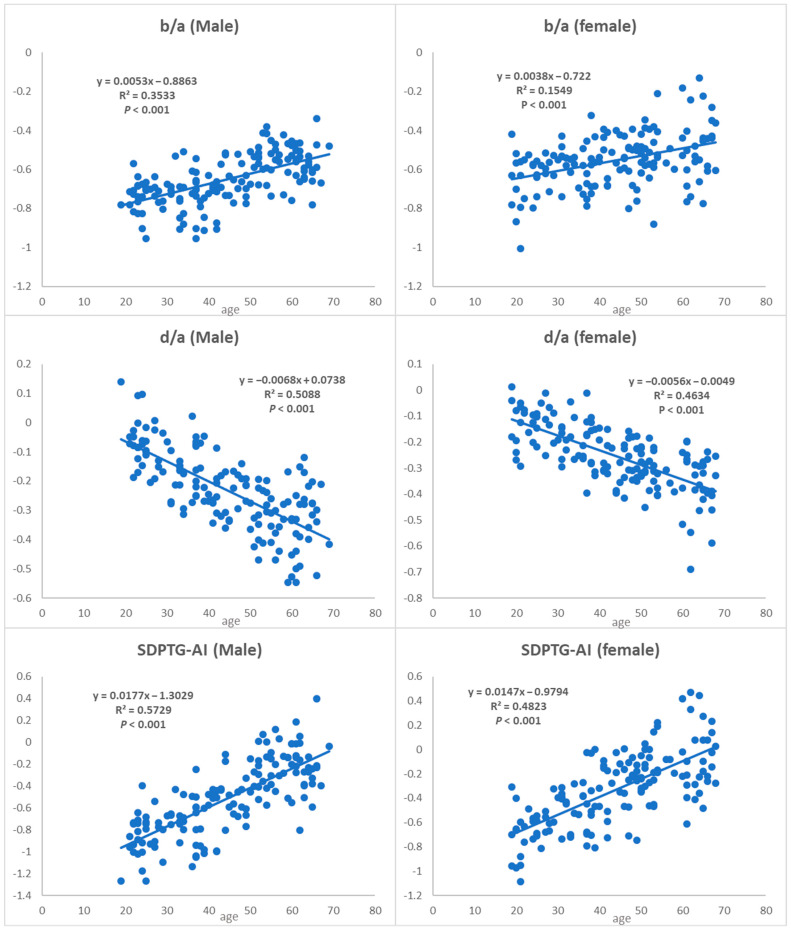
Variations in major second derivative photoplethysmogram (SDPTG) indices between male and female groups.

**Table 1 ijerph-20-00236-t001:** Comparison of second derivative photoplethysmogram (SDPTG) indices between male and female participants in each age group.

Variables	20s		30s		40s		50s		60s		Total	
	Male	Female	Male	Female	Male	Female	Male	Female	Male	Female	Male	Female
Subjects, *n*	30	29	30	29	27	30	30	27	30	30	147	145
Age, y	24.3 ± 2.5	23.5 ± 3.2	35.1 ± 2.7	35.3 ± 2.9	44.6 ± 2.9	45.3 ± 2.9	54.4 ± 2.7	52.4 ± 2.2 ^†^	63.1 ± 2.4	63.9 ± 2.6	44.3 ± 14.1	44.0 ± 14.3
b/a	−0.74 ± 0.08	−0.64 ± 0.13 ^†^	−0.74 ± 0.11	−0.59 ± 0.11 ^†^	−0.68 ± 0.1	−0.54 ± 0.11 *	−0.55 ± 0.09	−0.51 ± 0.12	−0.56 ± 0.09	−0.48 ± 0.17 ^‡^	−0.65 ± 0.13	−0.55 ± 0.14 ^‡^
c/a	−0.01 ± 0.1	0.01 ± 0.07	−0.01 ± 0.13	−0.08 ± 0.06	−0.08 ± 0.09	−0.13 ± 0.08 ^‡^	−0.16 ± 0.1	−0.16 ± 0.08	−0.16 ± 0.08	−0.16 ± 0.09	−0.08 ± 0.12	−0.10 ± 0.1
d/a	−0.07 ± 0.08	−0.13 ± 0.08 ^‡^	−0.17 ± 0.09	−0.2 ± 0.09 ^‡^	−0.24 ± 0.07	−0.28 ± 0.08	−0.33 ± 0.09	−0.31 ± 0.07	−0.34 ± 0.12	−0.36 ± 0.12	−0.23 ± 0.14	−0.25 ± 0.12
e/a	0.20 ± 0.1	0.14 ± 0.05 ^†^	0.17 ± 0.07	0.15 ± 0.05 ^†^	0.18 ± 0.05	0.15 ± 0.06 ^‡^	0.18 ± 0.07	0.11 ± 0.05 *	0.14 ± 0.06	0.11 ± 0.06 ^‡^	0.18 ± 0.07	0.13 ± 0.06 ^‡^
SDPTG-AI	−0.86 ± 0.2	−0.66 ± 0.18 *	−0.73 ± 0.20	−0.46 ± 0.22 *	−0.54 ± 0.2	−0.29 ± 0.21 *	−0.24 ± 0.19	−0.16 ± 0.19	−0.22 ± 0.24	−0.07 ± 0.28 ^‡^	−0.52 ± 0.33	−0.33 ± 0.3 ^‡^

Data are provided as number of participants or mean ± standard deviation (SD). *p*-values were derived using the independent *t*-test considering sex differences (males vs. females) in each age group. * *p* < 0.05, ^†^
*p* < 0.01, ^‡^
*p* < 0.001. b/a, ratio of the height of b wave to that of a wave of SDPTG; c/a, ratio of the height of c wave to that of a wave of SDPTG; d/a, ratio of the height of d wave to that of a wave of SDPTG; e/a, ratio of the height of e wave to that of a wave of SDPTG; SDPTG-AI, SDPTG aging index calculated as (b-c-d-e)/a. Age was divided into groups: 20s (19–29 years), 30s (30–39 years), 40s (40–49 years), 50s (50–59 years), and 60s (60–69 years).

## Data Availability

All the data supporting the findings of this study are available from the corresponding authors upon reasonable request.

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
