# Peer review of "Variations in the Second Derivative of a Photoplethysmogram with Age in Healthy Korean Adults"

_ijerph, 2022, doi:10.3390/ijerph20010236_

Round 1

Reviewer 1 Report

Comments.

This study is also the first to investigate all SDPTG indices for sex differences in the Korean population. The experiments were performed systematically, and data were interpreted methodologically. All the references are suitably cited. The work is of significance and meet the journal’s standards. However, there are certain concerns which need to be addressed before a favorable decision by the editors. The comments below listed below.

1.    The author didn’t mention, how long did they measure the PTG and second derivative PTG (SDPTG).

2.    The description of the different waves corresponding to the systolic and diastolic (a, b, c d and e) are insufficient in terms of their corresponding function. For instance, what does these waves signifies? What is the interpretation from b/a, c/a, d/a, e/a ratios? The authors should give a brief explanation about each wave or the ratios. This will help the readers to understand the correlation to the phenotype better. As in Figure 4 the authors has described b/a as the reflection of aortal stiffness, please describes each waves. Others have described c/a was related to heart rate variability (HRV) indices (Young-Jae Park., 2019). Please discuss

3.    The authors should present some representative data of the error in measurement which they observe during the recording?

4.    Figure 2 shows PTG and SDPTG signals measured for three participants of ages 20, 40, and 60 years. But the details about 30 and 50 years are missing?

5.    Simple regression models for SDPTG, hemodynamic, autonomic, and emotional factors to examine the effects of aging on the variables will be important. Furthermore, Pearson's correlation analyses between SDPTG and other variables would give more information and increase the significance of this study. At least they should discuss these variables and their prospects in this study.

Author Response

Please the attached file. 

Thank you very much for your valuable comments. 

Reviewer 2 Report

The authors performed analysis of variations in the second derivative of a photoplethysmogram (SDPTG) with age in healthy Korean adults. They defined sex-specific age-dependent functions for the maxima a, b, c, d, and e. Their main results are that (1) this is the first analysis of the SDPTG in Korean healthy subjects and that (2) age-dependent differences exist.

This is a well-written manuscript in good English with adequate statistics and citations. However, the value to readers/researchers outside of Korea is low as this study is subject to Korean people only. Furthermore, this study only includes healthy subjects. Consequently, no conclusions can be drawn for diseased people.

Minor comments:

-        In the conclusion, the authors write that these data may be valuable in preventing diseases related to vascular conditions by estimating the degradation of arterial function. They should better describe how their data is useful (e.g. defining standards for screening etc.).

Author Response

(The authors gave the same response as above.)

Round 2

Reviewer 1 Report

Thank You for addressing the previous comments. The responses are acceptable, and the manuscript has significantly improved. I have no further comments. I recommend adding a supplemental information on the patient information in terms of the inclusion criteria that is Please consider putting a supplemental table describing all the parameters listed in the manuscript method section (Participants were selected based on the following inclusion criteria 1 to 5 and the exclusion criteria 1 to 6) :)

Author Response

Dear reviewer,

Thank you for your valuable comment.

Detailed information on participants' SDPTG indices is attached to supplementary material.

Accordingly, a sentence is added in the Results section as follows.

"Detailed SDPTG indices on the participants are described in Table S1."